# Ocular Lymphatic and Glymphatic Systems: Implications for Retinal Health and Disease

**DOI:** 10.3390/ijms231710139

**Published:** 2022-09-04

**Authors:** Nasir Uddin, Matt Rutar

**Affiliations:** 1Centre for Research in Therapeutic Solutions, Faculty of Science and Technology, University of Canberra, Bruce, ACT 2617, Australia; 2Save Sight Institute, Faculty of Medicine and Health, The University of Sydney, Sydney, NSW 2000, Australia

**Keywords:** age-related macular degeneration, aqueous humour, glaucoma, glymphatic pathway, lymphatic system

## Abstract

Clearance of ocular fluid and metabolic waste is a critical function of the eye in health and disease. The eye has distinct fluid outflow pathways in both the anterior and posterior segments. Although the anterior outflow pathway is well characterized, little is known about posterior outflow routes. Recent studies suggest that lymphatic and glymphatic systems play an important role in the clearance of fluid and waste products from the posterior segment of the eye. The lymphatic system is a vascular network that runs parallel to the blood circulatory system. It plays an essential role in maintenance of fluid homeostasis and immune surveillance in the body. Recent studies have reported lymphatics in the cornea (under pathological conditions), ciliary body, choroid, and optic nerve meninges. The evidence of lymphatics in optic nerve meninges is, however, limited. An alternative lymphatic system termed the glymphatic system was recently discovered in the rodent eye and brain. This system is a glial cell-based perivascular network responsible for the clearance of interstitial fluid and metabolic waste. In this review, we will discuss our current knowledge of ocular lymphatic and glymphatic systems and their role in retinal degenerative diseases.

## 1. Introduction

The eye is a sensory organ responsible for the initial stages of visual transduction. It is filled with a transparent water-like fluid called aqueous humour within its anterior fluid compartments, which plays an essential role in the normal functioning of the eye. Dysfunction of the aqueous drainage system has been associated with a number of ocular disorders. It is generally considered that aqueous humour leaves the eye via two routes in the anterior eye, called the trabecular and uveoscleral pathways; these routes pass close to the lens and iris [1,2]. However, emerging evidence indicates that two other systems are also involved in aqueous humour outflow. The first of these is the lymphatic system, a vascular network that runs parallel to the blood circulatory system and drains excess fluid from most of the body organs [3]. The second is the more recently discovered glymphatic system, which is a glial cell-based waste-clearance network for the central nervous system [4].

The lymphatic and glymphatic systems have been suggested to play an important role in clearing interstitial fluid and metabolic waste from the retina, which is located in the posterior segment of the eye, as well as the optic nerve, which relays action potentials to the brain for processing. Despite their potentially important roles in fluid regulation, the presence and function of lymphatic and glymphatic systems in the eye are only just beginning to be unravelled. Improving our knowledge of ocular fluid drainage pathways is essential for better understanding of eye function in health and disease.

In this review, we will first provide an overview of ocular fluid aqueous humour, including its drainage routes and retinal fluid regulation pathways. We will then review current understanding of lymphatic and glymphatic systems and evidence for their presence in the eye. Finally, we will discuss the role of lymphatic and glymphatic systems in the pathogenesis of ocular disorders with particular focus on retinal degenerative diseases.

## 2. Aqueous Humour

Aqueous humour (AH) is a clear, slightly alkaline fluid contained in the anterior and posterior chambers of the eye, and is comprised of water, ions, carbohydrates, and proteins [1]. AH is primarily responsible for maintaining the intraocular pressure and shape of the eye and providing an optically clear medium to facilitate light transmission from the cornea to the retina. Moreover, AH provides nutrients and oxygen to and removes metabolic waste from avascular ocular tissues such as the cornea, lens, and trabecular meshwork [1,5,6]. It is similar to blood plasma in composition, though has a lower abundance of protein and glucose [7]. The protein component of AH includes mostly albumin with a small fraction of growth factors, and cytokines [8].

### 2.1. Production and Secretion

AH is continuously formed by the ciliary epithelium of the ciliary processes projecting from the pars plicata (anterior portion of the ciliary body) [9], at a typical rate of 2.5 μL/min or 3.6 mL per 24 h [10]. This formation process consists of three steps. Firstly, blood flows to the vascular bed of the ciliary processes. Secondly, plasma is filtered through the fenestrated ciliary capillaries into the ciliary stroma. Finally, the aqueous portion of the plasma is secreted by the ciliary epithelium as AH into the posterior chamber [9,11]. Three types of solute/liquid transport mechanisms are involved in AH formation: diffusion, ultrafiltration, and active transport [1,9].

Secretion of aqueous humour involves active transport of ions and other molecules through the non-pigmented epithelium of the ciliary epithelium into the posterior chamber, which in turn creates an osmotic gradient across non-pigmented epithelium (NPE) cells. This osmotic gradient then facilitates the passive water movement into the posterior chamber via aquaporin (AQP) water channels expressed by NPE cells: AQP1 and AQP4 [1,9,12]. After secretion into the posterior chamber, AH flows around the lens and enters the anterior chamber through the pupil [1].

### 2.2. Aqueous Humour Outflow

AH exits the eye through two pathways, called conventional and unconventional pathways (Figure 1). In the conventional pathway, also known as the trabecular pathway, AH passes through the trabecular meshwork, juxtacanalicular tissue, Schlemm’s canal, and collector channels and finally drains into the episcleral venous system [13,14]. This pathway is known as the primary route for AH drainage in human [15].

In the unconventional pathway, aqueous humour diffuses through the ciliary muscle, supraciliary (space between ciliary body and sclera), and suprachoroidal (space between choroid and sclera) spaces [2]. From there, AH leaves the eye via two routes: (1) AH flows across the sclera and drains into orbital blood vessels (uveoscleral pathway), or (2) AH enters the choroid and drains into vortex veins (uveovortex pathway) [2,14,17,18]. Moreover, recent studies have proposed a uveolymphatic pathway involving AH outflow through the lymphatic vessels in the ciliary body [19].

Although the bulk of AH exits the eye via trabecular and uveal routes, some portion of AH (~20%) flows posteriorly and enters the vitreous chamber in response to intraocular pressure [15,20,21]. As will be discussed in following section, AH then enters the retina and drains across the retinal pigment epithelium into choroidal blood circulation.

## 3. Fluid Regulation in Retina

The regulation of extracellular fluid is critical to retinal function in health and disease. Failure of retinal water regulation can cause retinal disorders such as retinal oedema [22]. In normal conditions, fluid accumulates in the retinal tissue due to (1) endogenous production of water because of metabolic activity, (2) influx of water from the blood associated with metabolic glucose and lactate uptake, and (3) water influx from the vitreous chamber in response to intraocular pressure (Figure 2) [21,23,24,25]. Clearance of this fluid from the retina is mediated by Müller cells and the retinal pigment epithelium (RPE). Müller glia eliminate water from the inner retina, whereas the RPE eliminates water from the sub-retinal space, situated between the photoreceptor cells and the RPE [23,26]. Moreover, water movement across the retina and out of the sub-retinal space is driven by intraocular pressure and choroidal osmotic pressure [21].

Müller glia remove water from the inner retina by facilitating water transport to the vitreous humour and retinal capillaries [26,27,28]. Transport of water through Müller cells is facilitated by the aquaporin-4 (AQP4) water channel protein, which is the predominant aquaporin in the mammalian retina [29,30]. AQP4 mediates rapid and bidirectional transport of water and small molecules across cell membrane [31]. The direction of the fluid transport depends on the trans-membranal osmotic gradient and on hydrostatic pressure [23,32]. In Müller cells, AQP4 is expressed on Müller cell endfeet, abutting the vitreous humour and on Müller cell side processes surrounding the blood capillaries [24,26].

Considerable evidence indicates that the water-transporting function of Müller cells is coupled to K^+^ currents that flow through these cells [24,27,33,34,35]. Under physiological conditions, neuronal activity in the retina results in the accumulation of extracellular K^+^ and water [23,36]. Müller cells clear excess K^+^ through their K^+^ channels from the retina to the vitreous body and blood; this process is known as K^+^ spatial buffering or K^+^ siphoning [27,37,38]. The predominant K^+^ channels in Müller cells responsible for K^+^ siphoning is the inwardly rectifying K^+^ channel 4.1 (Kir4.1) [37,39]. The Kir4.1 and AQP4 channels spatially colocalize on the Müller glial endfeet and perivascular processes [26,33]. This colocalization allows water to follow K^+^ from perisynaptic spaces to blood vessels [23,40,41].

Water from the subretinal space is cleared by transcellular ion and water fluxes through RPE cells [23]. In the human eye, RPE cells mediate subretinal fluid clearance via AQP4 water channels [42,43]. In contrast, in the rodent eye, where the RPE lacks AQP4 expression, transcellular fluid transport is facilitated primarily by cotransport proteins located at the apical and basolateral membranes of RPE cells [21,26,44].

## 4. Lymphatic System

The lymphatic system represents a second circulatory system, parallel to the blood vascular system. In humans, the lymphatic system is distributed throughout the body, with the exception of the central nervous system (CNS) [45]. The lymphatic system consists of a large network of lymphatic vessels and lymphoid organs (such as lymph nodes). The network of lymphatic vasculature is composed of initial lymphatic vessels (lymphatic capillaries) and collecting lymphatic vessels [45,46,47]. The primary functions of the lymphatic system include maintenance of tissue fluid homeostasis and immune surveillance [48].

### 4.1. Structure–Function Relationship of Lymphatic Vessels

The network of lymphatic vasculature is composed primarily of blind-ended and thin-walled capillaries. These lymphatic capillaries are lined by a single layer of overlapping endothelial cells that have a distinctive oak leaf-like shape [49]. Although lymphatic capillaries share many features with blood capillaries, they have very distinct structural and morphological features. The lymphatic capillaries have a more irregular and wider lumen compared to blood capillaries. Moreover, they lack a well-developed basement membrane and surrounding pericytes [48,50]. A hallmark feature of lymphatic capillaries is the presence of an anchoring filament by which lymphatic endothelial cells are connected to the surrounding extracellular matrix [46,51]. These overlapping lymphatic endothelial cells are loosely apposed and linked with each other by discontinuous button-like cell junctions [48,49]. In response to increased interstitial fluid pressure, the anchoring filaments pull open these intercellular junctions, allowing interstitial fluid and cells into the capillary lumen. Following fluid entry, the cell junctions close and prevent the retrograde flow of fluid into interstitial space [46,48,49,52].

In contrast to lymphatic capillaries, collecting lymphatic vessels are made up of spindle-shaped endothelial cells. These vessels have a complete basement membrane and are encircled by one or two layers of smooth muscle cells [49,53,54]. Moreover, they have intraluminal valve structures to ensure the unidirectional flow of lymph fluid. Lymphatic endothelial cells of the collecting vessels are connected to each other by continuous zipper-like junctions [49,55].

Lymphatic vessels maintain tissue fluid homeostasis by clearing interstitial fluid. At increased interstitial fluid pressure, lymphatic capillaries collect excess extracellular fluid and pass this fluid, now called lymph, to collecting lymphatic vessels. From these vessels, lymph then flows into and out of lymph nodes via afferent and efferent lymphatic vessels, respectively. Finally, lymph drains into blood circulation through the lymphaticovenous junction between the thoracic or lymphatic duct and the subclavian veins [45,49,56,57].

### 4.2. Functions of the Lymphatic System

The lymphatic system performs several important functions, including tissue fluid balance, dietary fat absorption, and immune surveillance [50]. Lymphatic capillaries collect fluid and macromolecules extravasated from blood vessels and return to blood circulation by collecting lymphatic vessels, thus preventing excess build-up of fluid in tissues (oedema) [46,58]. The lymphatic vasculature also serves as a major route for absorption of fats and fat-soluble vitamins from the digestive tract [58]. They also transport and remove waste products, cell debris, virus, and bacteria, from the lymph in the lymph node. By transporting leukocytes, including T and B lymphocytes, as well as antigen-presenting cells from tissues to lymph nodes, lymphatics play a critical role in initiating the immune response [46,59]. Lymphatics are also involved in tumour metastasis [60] and inflammatory response regulation [61]. Besides, lymphatic endothelial cells have been reported to play an essential role in cardiac growth and repair [62] as well as in intestinal repair [63] via lymphangiocrine signalling.

### 4.3. Challenges in Identification of Lymphatic Vessels

Despite the availability of several antigenic markers, unequivocal identification of lymphatic vasculature remains challenging, particularly in the eye. Lymphatic capillaries are often irregular and collapsed, thus making it difficult to visualize them histologically [58]. Moreover, until recently, lymphatics were identified based on limited histological criteria, including the absence of luminal red blood cells, and well-developed basement membrane [64]. With these criteria, lymphatic vessels were difficult to distinguish definitively from blood vessels. To facilitate histological identification of lymphatics, recent studies have reported several immunomarkers: podoplanin, lymphatic vessel endothelial hyaluronic acid receptor-1 (LYVE1), prospero-related homeobox-1 (PROX-1), and vascular endothelial growth factor receptor-3 (VEGFR-3) [65]. Most of these proteins play important roles in the development of lymphatic vessels and in the formation of new lymphatic vessels (lymphangiogenesis) [65]. Among these widely used markers, however, no single marker is exclusively specific to lymphatics. Therefore, the following criteria have been recommended for unequivocal immunohistochemical identification of lymphatic vasculature: (1) multi-maker immunohistochemistry including at least two lymphatic endothelial markers, and (2) use of appropriate control tissue for lymphatic markers [66].

### 4.4. Lymphatics in the Eye

The eye was long considered to lack lymphatics except for the conjunctiva [67]. Previous studies also reported lymphatic vessels in the corneal limbus [68,69]. Under normal conditions, the cornea is devoid of any vasculature including lymphatics. However, in pathological conditions such as inflammatory corneal neovascularization, lymphangiogenesis is induced in the limbus region (corneal–conjunctival border) of the cornea, and formation of these lymphatic vessels is independent of blood vessel formation (angiogenesis) [70,71,72]. A recent study reported that distribution of limbal and conjunctival lymphatics is highly polarized toward the nasal side of the eye [73]. Lymphatic vessels were also reported in the ciliary body [19]. Following immunohistochemical studies on post-mortem human eyes, Yücel et al. found podoplanin and LYVE1-positive lymphatic channels in the ciliary body stroma. Moreover, the authors found that ciliary body lymphatics were involved in drainage of aqueous humour from the anterior chamber of the eye [19]. The presence of ciliary body lymphatics has also been investigated by several other groups; however, the results were conflicting. One consistent finding of these studies was the presence of LYVE1-positive cells in the ciliary body stroma and processes [74,75,76,77]. Interestingly, the majority of these cells was also immunoreactive for CD68 (a well-known macrophage marker), indicating that these cells might belong to the macrophage lineage [75]. However, this does not imply that the classical lymphatics are indeed absent in the ciliary body. Instead, it was suggested that ciliary-body lymphatics might express different unknown molecular markers, as reported for Schlemm’s canal [75,78]. Therefore, further studies are needed to confirm the presence or absence of ciliary-body lymphatics in both physiological and pathological conditions.

In the posterior part of the eye, lymphatic capillaries have been identified in the developing and adult human choroid [50]. However, in the choroid, especially in the healthy adult human choroid, controversy still remains regarding the presence of classical lymphatics similar to those in the ciliary body [79,80,81,82]. Although there is to date no report of traditional lymphatic vasculature in the retina and sclera, a large population of non-endothelial LYVE1-positive macrophages was reported in the sclera [72,83,84,85]. Furthermore, lymphatic capillaries have been identified in human optic nerve meninges, but evidence of these lymphatics is limited. These limitations are mainly attributable to the use of inadequate immunomarkers and ultrastructural criteria to detect lymphatic vessels [86,87,88]. Meningeal lymphatics have been suggested to play an important role in the drainage of cerebrospinal fluid [87,89], but it is not clear whether they are also involved in aqueous humour outflow from the eye.

In summary, it appears that the presence of ocular lymphatics, particularly in the uvea, remains highly debated. Based on the critical literature review, it is clear that most of the studies on ocular lymphatics did not use (1) both the structural and functional experimental approaches or (2) both the multiplex immunolabeling and ultrastructural (including immunogold labelling) studies. Therefore, future studies could be extended to multi-modal experimental approaches to confirm the presence or absence of ocular lymphatics.

## 5. Glymphatic System

Although CNS has long been thought to lack a classical lymphatic vasculature, recent studies have discovered a quasi-lymphatic system (termed the glymphatic system) in the brain parenchyma [4], as well as a traditional lymphatic system in the meninges that envelop the brain [90]. The glymphatic (glial-lymphatic) system is a glial cell-dependent perivascular network that subserves a lymphatic-like function by clearing metabolic waste from the brain parenchyma [4,91]. The glymphatic waste-clearance pathway consists of three steps: (1) from the sub-arachnoid space, cerebrospinal fluid (CSF) flows into the periarterial space, which is the perivascular space around the arteries penetrating the brain. From there, CSF enters the brain parenchyma via the astrocytic water channel AQP4 and mixes with interstitial fluid (ISF). (2) As CSF exchanges with ISF, vectorial convective fluxes drive metabolic waste products away from the arteries and towards the vein. (3) ISF then leaves the brain along the perivenous space, which is the perivascular space surrounding large-calibre draining veins [4,91,92,93,94] (Figure 3). Previous studies have already reported perivascular transport of CSF and its involvement in waste clearance from the brain [95,96,97,98,99,100]. The astrocytic aquaporin channel AQP4 also participates in perivascular CSF–ISF exchange across brain parenchyma [4,101]. Beyond clearance, the glymphatic pathway is also involved in brain-wide distribution of nutrients, including glucose [102], and astrocytic paracrine signalling [103,104]. Dysfunction of the glymphatic system is associated with the pathophysiology of a variety of neurological disorders such as Alzheimer’s disease [105], traumatic brain injury [106], stroke [107], sub-arachnoid haemorrhage [108], and multiple sclerosis [101].

### 5.1. Components of the Glymphatic System

#### 5.1.1. Cerebrospinal Fluid

Cerebrospinal fluid (CSF) is a clear, proteinaceous fluid that occupies the ventricles of the brain and the sub-arachnoid space surrounding the brain, spinal cord, and optic nerve. The CSF volume in adult human is estimated to be 150 mL, with 125 mL in the sub-arachnoid space and 25 mL in the ventricles [109]. The CSF is produced primarily by the choroid plexus located in the lateral, third, and fourth cerebral ventricles [31]. Choroidal production and secretion of CSF involves two steps: (1) a pressure-driven passive filtration of plasma from the choroidal capillaries to the choroidal interstitial compartment, and (2) an active transport from the interstitial compartment to the ventricles across the choroidal epithelium, controlled by carbonic anhydrase and membrane ion-carrier proteins [109,110]. Following production in the lateral ventricles, CSF flows through the third and fourth ventricles linked by channels or foramina into the sub-arachnoid space of the cortex, spinal cord, and optic nerve. From the cortical sub-arachnoid space, CSF flows into and out of the brain parenchyma via the glymphatic pathway [91,101]. Throughout the circulation, CSF moves by convective flow, which is driven by arterial pulsation and respiration [111,112]. The CSF exits the brain and ultimately drains into the bloodstream via two major pathways: (1) through arachnoid granulations of the dural sinuses to the blood, and (2) along cranial nerve sheaths to the cervical lymphatic system [101,113,114]. Emerging evidence shows meningeal lymphatics lining the dural sinuses are also able to collect CSF from the sub-arachnoid space and drain into deep cervical lymph nodes [90,115].

#### 5.1.2. Cerebral Blood Vessels and Perivascular Space

Pial arteries and veins run through the sub-arachnoid space and are surrounded by CSF. Among these blood vessels, only pial arteries are enveloped by pia mater [116]. These arteries penetrate the brain parenchyma and branch into arterioles and capillaries. Parenchymal arterioles are also covered by pia mater for a short distance; however, this covering disappears at the capillary level [91,117]. All small blood vessels (arterioles, capillaries, and venules) within the brain parenchyma are bordered by astrocytic endfeet [91].

The perivascular space is a fluid-filled compartment that surrounds the blood vessels (arteries and veins), penetrating into the brain parenchyma [118]. This space is also known as paravascular space or Virchow–Robin space [4,31,91,118]. Perivascular space is generally bound by the abluminal surface of blood vessels anteriorly and astrocytic endfeet exteriorly [94]. However, the perivascular space around penetrating arteries is also accompanied by pia mater on both the inner and outer wall-facing vessels and astrocytes, respectively [91,117]. As the penetrating arteries branch into parenchymal arterioles, the perivascular space narrows and finally disappears at the capillary level [91]. In the past, perivascular space was considered a histopathological fixation artefact due to the unavailability of reliable techniques for its identification [117,119]. Recent developments of advanced in vivo imaging techniques such as two-photon imaging and magnetic resonance imaging (MRI) enabled detection of perivascular space in the rodent brain [114,120], and perivascular space is also detectible in the human brain with high-resolution MRI [121].

#### 5.1.3. Astrocytic Water Channel Aquaporin-4

AQP4 is the predominant water channel protein in the brain, where it is expressed by astrocytes and ependymal cells [122]. Astroglial AQP4 plays an important role in trans-parenchymal fluid transport pathway in the brain. In the first study on the glymphatic system, Illif et al. proposed that AQP4 facilitates the convective bulk flow of fluid along perivascular compartments and through the brain interstitium [4]. To test this hypothesis, they assessed CSF transport across the brain parenchyma in global *AQP4* knockout mice. Using in vivo two-photon and ex vivo fluorescence imaging, they found that CSF transport into and through the brain interstitium is significantly reduced in *AQP4*^−/−^ mice compared to wild-type control mice. Moreover, they observed that interstitial solute clearance from the brain was reduced by ~70% in *AQP4*^−/−^ mice [4,106]. The importance of AQP4 for glymphatic flow has also been supported by other studies carried out in both global and glial-conditional *AQP4* knockout mice [123,124,125,126]. Polarized expression of astrocytic AQP4 plays a crucial role in the glymphatic pathway. Loss of perivascular AQP4 localization results in impaired perivascular fluid transport and interstitial waste clearance, as observed in both the young and aging mouse brain following traumatic brain injury [101,120,126,127].

#### 5.1.4. Ocular Glymphatic System

As part of the CNS, the eye—particularly the retina and optic nerve—shares embryological, anatomical, and physiological similarities to the brain [128,129]. In common with the brain, both the retina and optic nerve parenchyma lack a lymphatic system [67], suggesting that they might possess an alternative system for fluid homeostasis and waste clearance. The high metabolic activity of retinal neuron resulting in the generation of neurotoxic waste products also supports the notion that the retina might have an alternative waste-clearance pathway [77]. Moreover, the retina and optic nerve also have astrocytes and AQP4 water channels on the astrocytic endfeet [30,130,131]. Considering this, several hypothesis-driven studies have reported the presence of a glymphatic-like system in the eye [132,133,134,135,136,137]. In 2015, the Denniston [132] and Wostyn [137] research groups independently hypothesized the existence of a glymphatic-like system in the retina and optic nerve, which might play an important role in pathophysiology of retinal diseases such as age-related macular degeneration and glaucoma. However, to date there are only two experimental studies reporting the presence of an ocular glymphatic system (summarized in Figure 4) [89,138]. The first evidence was reported by Mathieu et al. who, examined the route of CSF influx into the mouse optic nerve [138]. In this study, the authors injected dextran tracers into CSF-filled cisterna magna, which is an opening located in meninges enveloping the brain. One hour after tracer injection, the authors found that CSF entered the optic nerve parenchyma via perivascular space around small penetrating pial vessels, suggesting the presence of a glymphatic pathway in the optic nerve [138]. However, the authors acknowledged that their study was limited to the assessment of a single time point, due to which it was not possible to evaluate differential entry and exit routes of CSF into the optic nerve [138]. Further studies with different time points are needed to fully characterize the optic nerve glymphatic system. The presence of an ocular glymphatic system in the mouse eye was also recently reported by Wang et al. [89]. Following intravitreal injection of human amyloid beta tracers into mouse eyes, Wang et al. observed that these tracers were taken up by retinal ganglion cells and cleared from the retina along the optic nerve via astroglial AQP4 water channels [89]. Beyond the optic nerve head, intra-axonal tracers were cleared via the perivenous space and subsequently drained into meningeal lymphatic vessels [89]. Moreover, translaminar pressure difference was responsible as a driving force for the glymphatic clearance in murine eyes, and this clearance was further increased due to pupil constriction in response to light [89]. However, the authors raised several questions in their study: (1) whether the ocular glymphatic system is the principal waste-clearance route for retinal ganglion cells, and (2) whether and how different subtypes of retinal ganglion cells influence this clearance pathway [89]. Moreover, from their study, it is not clear whether Müller glia are also responsible for the uptake of tracers from the vitreous chamber, given the attachment between vitreous and Müller glia endfeet at the vitreoretinal border [32] To date, all experimental studies on CNS and ocular glymphatic systems have been carried out on rodents. Since rodents and primates, including humans, have cellular and anatomical differences in the retina and optic nerve of the eye, further research is needed to characterize the glymphatic system across mammalian species.

## 6. Lymphatics/Glymphatics in Retinal Degenerative Diseases

Growing evidence suggests that ocular lymphatic and glymphatic systems are involved in the pathogenesis of a number of disorders associated with ocular fluid homeostasis and waste clearance. The following section reviews current research investigating the role of lymphatic/glymphatic drainage in retinal diseases.

### 6.1. Glaucoma

Glaucoma is a multifactorial optic neuropathy characterized by the slow progressive degeneration of retinal ganglion cells (RGCs) and their axons within the optic nerve. This degeneration results in structural changes in the optic nerve head and subsequent visual field defects, leading to blindness [16,139]. It is recognized that elevated IOP plays an important role in glaucomatous optic nerve degeneration [16].

The lymphatic system has been suggested to play a role in glaucoma pathogenesis due to its involvement in aqueous humour drainage. One recent study showed that angiopoietin (ANGPT) growth factors are critical for the development of lymphatic vessels in the corneal limbus, and the loss of ANGPT1 and ANGPT2 from lymphatics is associated with glaucoma and ocular hypertension in mice [140]. Impaired function of ciliary-body lymphatics has also been proposed to be associated with glaucoma, but no experimental evidence is available in support of this [141]. Lymphatics located in optic nerve meninges have been reported to play an essential role in the drainage of CSF [87,89]. In a study carried out on postmortem human eyes, Killer et al. injected India ink into the subarachnoid space of the optic nerve meninges to track the CSF drainage route. Following injection, the authors located India ink particles within the lumen of lymphatic capillaries in the dura mater (the outermost layer of the meninges), suggesting the possible role of meningeal lymphatics in CSF drainage function [87]. Lymphatics in brain meninges are also involved in CSF clearance, and their dysfunction has been associated with a number of neurodegenerative diseases [115,142]. In a recent study, Aspelund et al. demonstrated that lymphatics located in the dura mater of the mouse brain collect CSF from adjacent subarachnoid space and drain into deep cervical lymph nodes [115]. In a transgenic mouse model, the authors also showed that absence of the dural lymphatics was responsible for the delayed clearance of macromolecules injected into mouse brain parenchyma, suggesting the potential role of meningeal lymphatics in CNS disorders such as Alzheimer’s disease [115]. However, it is not clear whether meningeal lymphatics in the optic nerve play such a role in ocular neurodegenerative disorders such as glaucoma.

The glymphatic system has also been implicated in the pathophysiology of glaucoma. Following intravitreal injection of human amyloid beta in glaucomatous rodent eyes, glymphatic clearance of amyloid beta from the retina decreased significantly [89]. Moreover, in a tracer-based study, Mathieu et al. showed that glymphatic entry of CSF into the mouse optic nerve is reduced in glaucoma [143].

### 6.2. Age-Related Macular Degerations

Age-related macular degeneration (AMD) is a retinal disease associated with aging that gradually affects the macula—a specialised region in the retina responsible for sharp central vision needed for simple everyday activities [144]. The pathogenesis of AMD is multifactorial, including the effects of aging and oxidative stress as well as genetic and environmental factors [145]. Evidence of the significance of immune response in AMD pathogenesis, including infiltration of inflammatory cells, has also been accruing for some time [146,147].

The ocular glymphatic system has been hypothesized to be involved in AMD pathogenesis. This hypothesis is based on the facts that (1) the glymphatic pathway clears metabolic waste, including amyloid beta peptide, from the CNS and AMD is associated with amyloid beta deposition [136,148], and (2) glymphatic transport of CNS waste is impaired significantly with ageing, which is a risk factor for AMD development [120,136]. However, there is no direct research evidence available in support of glymphatics’ role in AMD pathology.

### 6.3. Other Retinal Diseases

Recent studies suggest that an impaired ocular lymphatic system due to ageing might be associated with ocular inflammatory diseases such as uveitis [149,150]. However, further experiments are needed to better understand the role of the ocular lymphatic system in the pathophysiology of uveitis. Macular oedema has been hypothesized to be associated with ocular glymphatic dysfunction [133,151], although experimental works are yet to be carried out in support of this.

## 7. Potential Avenues for Lymphatic/Glymphatic Modulation

Emerging evidence indicates that physiological, pharmacological, and lifestyle interventions may help improve CNS glymphatic function under physiological and pathological conditions. Over the past few years, natural sleep has been implicated in glymphatic function under physiological conditions. In a recent study on a rodent model, Xie et al. found that natural sleep significantly increased CSF–ISF exchange in mouse brain parenchyma with a consequent increase in the glymphatic clearance of amyloid beta [152]. Under pathological conditions such as dementia, sleep disturbances have been found to delay glymphatic clearance of metabolic waste from the CNS [153].

Given that CNS glymphatic function is facilitated by astrocytic AQP4 water channels, a number of AQP4 inhibitory and facilitatory compounds have been investigated for their modulatory effects on glymphatic function. In a rat stroke model, AER-271 (a potential AQP4 inhibitor) has been found to reduce cerebral oedema, suggesting that this compound might be used to improve the glymphatic function under neurological conditions. This suggestion was based on the fact that glymphatic system has been reported to be associated with cerebral oedema formation following ischemic stroke [154]. Other AQP4 inhibitors such as AER270 and IMD-0354 have also been found to alleviate cerebral oedema [154,155]. Besides the AQP4 inhibitors, AQP4 facilitator compounds have also been identified. One such compound is TGN-073, which binds to AQP4 channels, leading to a conformational shift and increase water flux, resulting in increased glymphatic function across the brain parenchyma [156,157].

A recent study on a rodent model of Alzheimer’s disease was carried out to investigate the relationship between exercise (a lifestyle factor) and glymphatic function [156,158]. The results showed that voluntary exercise increased glymphatic influx of CSF tracers in the brain of awake, behaving young mice [158]. Likewise, voluntary running was also found to upregulate astrocytic AQP4 expression, leading to increased glymphatic activity [159].

In contrast to the CNS glymphatic system, limited studies have been carried out investigating the modulation of similar systems in the eye. In a recent study, Wang et al. demonstrated that light-induced pupil constriction increased the glymphatic clearance of intravitreally injected human amyloid beta from mouse retina [89]. Moreover, this increase was eliminated when pupillary light reflex was blocked with topical administration of atropine on mouse eyes [89]. However, further studies could be carried out to investigate the role of other physiological, pharmacological factors in ocular glymphatic function.

Several recent studies reported the potential of lymphatic modulation in the treatment of retinal diseases. Regulation of lymphatic trafficking has been reported to promote corneal graft survival [160]. A variety of therapeutic strategies including pharmacological interventions have been developed to target the inhibition of lymphangiogenesis to improve corneal transplant survival. For example, anti-lymphangiogenic agents VEGFRs, VEGFR-tyrosine kinase inhibitors, thrombospondin-1, and integrin-blocking peptides have been tested and reported to be efficient in improving corneal surgery outcome in an experimental setting [160,161,162,163]. A recent study reported that adrenergic glaucoma drugs such as timolol and betaxolol play an important role in the modulation of ocular lymphatic drainage [164]. Latanoprost, which is also a widely used glaucoma drug, has been reported to reduce intraocular pressure by increasing aqueous humour outflow via uveoscleral pathway [165]. However, it is not clear whether latanoprost is also involved in the regulation of uveal lymphatic clearance. Future studies could be carried out to better understand different modulators of ocular lymphatics and their role in the pathophysiology of ocular diseases.

## 8. Conclusions

Over the last few years, substantial efforts have been made to understand ocular lymphatic and glymphatic systems, with a particular focus on their role in ocular pathophysiology. Findings from these studies contributed to our understanding of ocular fluid-drainage routes. It is generally considered that ocular fluid, particularly aqueous humour, drains via trabecular and uveoscleral pathways from the anterior segment of the eye [2]. Previous studies on ciliary body lymphatics showed that this lymphatic system also contributes to the anterior outflow pathway of aqueous humour [19]; however, more studies are needed to better understand the role of lymphatics in ocular fluid regulation. Little is known about the clearance of interstitial fluid, including metabolic waste, from the retina. Recent evidence of the ocular glymphatic system showed that this alternative lymphatic system plays an important role in fluid homeostasis and metabolic waste clearance in the posterior part of the eye, particularly in the retina and optic nerve [89,138]. Moreover, studies on ocular lymphatic and glymphatic systems showed that these systems suggest an association with the pathophysiology of several prominent retinal degenerative diseases, including age-related macular degeneration [136]. Although further investigation is needed, our burgeoning understanding of ocular lymphatic and glymphatic systems shows clear potential to reveal new insights into ocular health and disease. In conjunction, our growing understanding of the factors that modulate ocular lymphatic/glymphatic drainage could serve as an important tool in ameliorating retinopathy and optic neuropathy in the future, potentially leading to improved vision health outcomes.

## Figures and Tables

**Figure 1 ijms-23-10139-f001:**
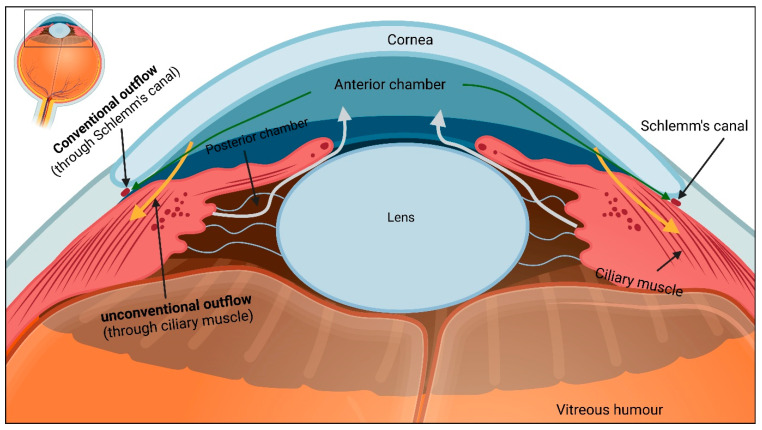
Aqueous humour outflow pathway. After secretion into the posterior chamber, aqueous humour enters the anterior chamber (light grey arrows) and leaves the eye through the trabecular meshwork (green arrows) and uveoscleral route (orange arrows). Figure inspired from [16].

**Figure 2 ijms-23-10139-f002:**
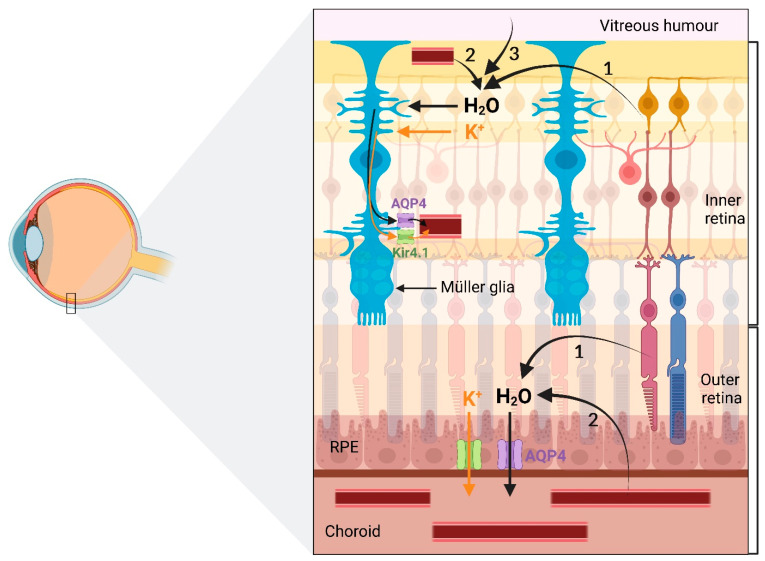
Retinal fluid regulation pathway. Under normal conditions, water accumulates in the retina due to (1) metabolic activity, (2) influx from blood, and (3) influx from the vitreous chamber. Müller glia and RPE cells are responsible for water clearance from the inner and outer retina, respectively. This transcellular water transport is facilitated by the AQP4 water channel and is coupled to transport of potassium ions (K^+^) through the K^+^ channel. Figure inspired from [25].

**Figure 3 ijms-23-10139-f003:**
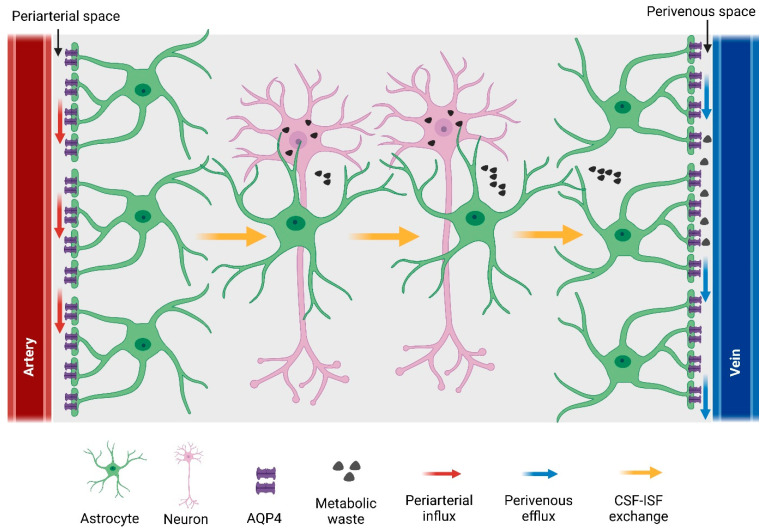
CNS glymphatic pathway. CSF enters the brain parenchyma along the periarterial space, clears the waste products from the brain, and leaves the brain along the perivenous space. CSF flow into and through the brain interstitium is facilitated by astrocytic AQP4 water channels.

**Figure 4 ijms-23-10139-f004:**
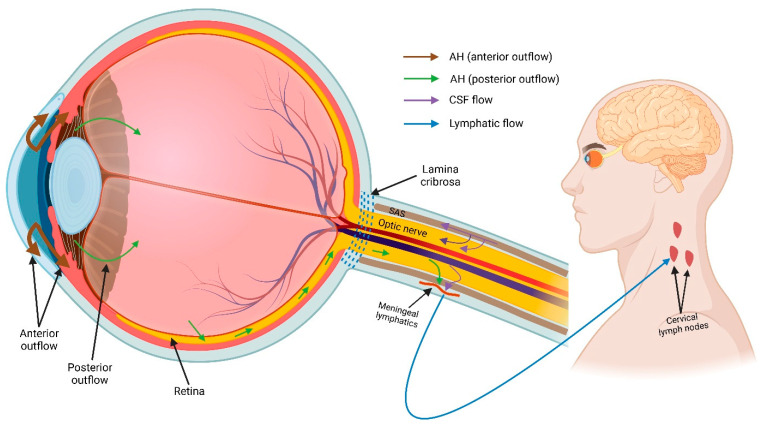
Ocular glymphatic system. Schematic showing the current understanding of the ocular glymphatic system. Following production, a major portion of AH leaves the eye via the anterior outflow pathway (brown arrows). A small portion of AH enters the vitreous chamber (green arrows), from where AH enters the neural retina due to intraocular pressure. In the retina, AH mixes with interstitial fluid and transport along retinal ganglion cell axons across the lamina cribrosa barrier. From there, AH leaves the axons, travel towards the perivenous space, and finally drains into cervical lymph nodes (blue arrow) via meningeal lymphatic vessels. Likewise, the optic nerve has a distinct glymphatic system. In this pathway, CSF from the sub-arachnoid space (SAS) enters the optic nerve parenchyma (purple arrows) along the periarterial space. Following glymphatic activity, CSF then leaves the optic nerve via perivenous space before draining into cervical lymph nodes via meningeal lymphatics.

## Data Availability

Not applicable.

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
