# Peer review of "Ocular Lymphatic and Glymphatic Systems: Implications for Retinal Health and Disease"

_ijms, 2022, doi:10.3390/ijms231710139_

Round 1

Reviewer 1 Report

General Comments :

The objective of this review is to report and discuss the current knowledge concerning the ocular lymphatic and glymphatic systems, with their potential repercussions in retinal disorders. After a general presentation of ocular fluids dynamics, and of both lymphatic and glymphatic systems, the authors reported data concerning these systems in the eye and discussed some of their potential implications in ocular disorders.

The topic of the review is attractive. Indeed, there are still several unsolved questions remaining concerning the importance of both lymphatic and glymphatic systems in the eye. These two systems could have implications for the development of novel therapeutic strategies, especially concerning neurodegenerative eye diseases. Regular updates and new hypothesis would then be beneficial and deserve to be regularly done. However, the reader could be disappointed by the content of the review in its present form since it appears unbalanced and does not add much to the few recently published reviews on the subject. A large part of its content is devoted to the general presentation of both lymphatic and glymphatic systems, whereas the information concerning the eye lymphatics and the eye glymphatics are not sufficiently developed.

Some major modifications would then be necessary to improve the manuscript and to complement the existing literature on the subject. Overall, a better focus on the ocular lymphatics and ocular glymphatics would be preferable. The available information should be more fully presented and further discussed to provide a more comprehensive review. Some recent work on the distribution and the organogenesis of ocular lymphatics should be included. The existing controversy concerning the existence of lymphatics in the internal part of the eye should be presented. A figure illustrating the putative retinal glymphatics would be bring more than Figure 2 which illustrates the well-described glymphatic system of the brain CNS. Moreover, Figure 1 should be corrected (see specific comments).

Specific Comments :

11) In the Abstract, the statement that lymphatic vessels have recently been described in the cornea should precise that this is occuring only in response to inflammation. As stated by the authors in paragraph 4.4, the cornea is normally avascular. The last sentence of the abstract should be restrictive to retinal diseases instead of ocular disorders, in order to match with the title of the review.

22)  In the Introduction, first paragraph, the reference [3] is not really suitable to the presentation of the lymphatic system. It should be replaced by a reference of a more general review from one of the group leader in the lymphatic research domain. This is also the case in paragraph 4  for the general presentation of the lymphatic system. Reference [47], dedicated to the study of the potential existence of lymphatics in the choroid, is not suitable for introducing the primary functions of the lymphatic system. It would be preferable to cite relevant major publications in the domain of interest.

33) Figure 1 needs corrections. The modifications introduced from the figures appearing in reference [16] are not all correct. The illustrated position of the posterior chamber is not right, since it should be located below the iris. Similarly, the picture does not corresponds to the text with regard to the position of the posterior chamber. On the other hand, the Schlemm’s canal is located at the iridocorneal angle and, in my opinion, does not constitute part of the cornea. Finally, the iris is apposed to the lens in the case of Closed angle glaucoma, but not in healthy conditions.

44) A figure illustrating fluid regulation in the retina would be helpful.

55) The paragraph 4.2 « Functions of the lymphatic system » describes the traditional well-known functions of the lymphatic system. The recently reported novel functional roles of the lymphatic vasculature with lymphangiocrine signalling should be briefly introduced.

66)  Paragraph 4.3 « Identification of lymphatic vessels » should be revised. It’s more than two decades since antigenic markers which discriminate lymphatic from blood endothelial cells have been characterized. Moreover, the sentence indicating that « these proteins play important roles in the development of lymphatic vessels » is not adapted for LYVE-1. To my knowledge, the role of LYVE-1 in lymphatic development remains unclear since LYVE-1-deficient mice display a normal lymphatic vasculature.

77) Paragraph 4.4 « Lymphatics in the eye » should be extended to report and discuss the remaining controversy concerning the presence of true classical lymphatics in uveal tissues and in the posterior eye. It should also be specified that the non-endothelial LYVE-1 positive cells belong to the macrophage lineage [reference 69]. The recent litterature on the ocular lymphatic vessel distribution and organogenesis should be included.

88) Paragraph 7 « Potential avenues for lymphatic/glymphatic modulation » mainly deals with the glymphatic system of the CNS. The reader would expect more details and/or hypothesis concerning the glymphatics of the eye. It would also be important to discuss the perspectives for the modulation of ocular lymphatics.

99)  In the conclusion, the statement that the ocular glymphatic system has been studied thoroughly seems somewhat overstated since in the previous paragraph 5.1.4, the authors wrote that only 2 studies reported the existence of an ocular glymphatic system. On the other hand, the importance of lymphatics in IOP regulation remains, in my opinion, to be fully established. The authors should take into account these remarks and revised the text of the conclusion, accordingly.

Reviewer 2 Report

This review presented the lymphatic and glymphatic systems in the eye. They introduced fluid clear systems, including aqueous humour, lymphatic, and glymphatic system in the pathogenesis of the ocular disease, especially focusing on retinal degenerative diseases. The authors summarized novel insights of ocular disease pathogenesis which provide a new direction to both uncover the mechanism of said disease and develop new treatments to improve patient vision. The authors explained their points clearly and thoroughly with appropriate citations. However, if the authors could provide one or two schematic figures that illustrate the function of the lymphatic and glymphatic systems in healthy and diseased eye conditions, that would help readers better understand the areas of interest.

Round 2

Reviewer 1 Report

All the previous comments and suggestions have been addressed satisfactorily.

Moreover, the manuscript content has been greatly improved by the addition of new figures supporting the text. 

I have only a suggestion for a minor modification that the authors could incorporate on the new Figure 4 (p11). Since the major Aqueous Humor (AH) outflow occurs through the anterior part of the eye and since only a smaller portion enters the vitreous chamber, as specified in the figure legend, the thickness of the corresponding green arrows could be adjusted to be consistent with this statement. Arrows of different colors could be useful to distinguish anterior from posterior AH outflows, instead.

Author Response

Response: We thank reviewer for this suggestion, and in response we have modified the figure and revised relevant text of figure legend as follows (section 5.1.4, page 19):